# Quaternized Curcumin Derivative—Synthesis, Physicochemical Characteristics, and Photocytotoxicity, Including Antibacterial Activity after Irradiation with Blue Light

**DOI:** 10.3390/molecules29194536

**Published:** 2024-09-24

**Authors:** Pawel Bakun, Marcin Wysocki, Magdalena Stachowiak, Marika Musielak, Jolanta Dlugaszewska, Dariusz T. Mlynarczyk, Lukasz Sobotta, Wiktoria M. Suchorska, Tomasz Goslinski

**Affiliations:** 1Chair and Department of Chemical Technology of Drugs, Poznan University of Medical Sciences, Rokietnicka 3, 60-806 Poznan, Poland; magdalena.m.stachowiak@gmail.com (M.S.); mlynarczykd@ump.edu.pl (D.T.M.); 2Doctoral School, Poznan University of Medical Sciences, Bukowska 70, 60-812 Poznan, Poland; marcin.wysocki@student.ump.edu.pl (M.W.); marika.musielak@wco.pl (M.M.); 3Chair and Department of Inorganic and Analytical Chemistry, Poznan University of Medical Sciences, Rokietnicka 3, 60-806 Poznan, Poland; lsobotta@ump.edu.pl; 4Department of Electroradiology, Poznan University of Medical Sciences, Garbary 15, 61-866 Poznan, Poland; wiktoriasuchorska@ump.edu.pl; 5Radiobiology Laboratory, Department of Medical Physics, Greater Poland Cancer Centre, Garbary 15, 61-866 Poznan, Poland; 6Chair and Department of Genetics and Pharmaceutical Microbiology, Poznan University of Medical Sciences, Rokietnicka 3, 60-806 Poznan, Poland; jdlugasz@ump.edu.pl

**Keywords:** curcumin derivatives, PACT, Microtox, morpholine, MTT, toxicity

## Abstract

Over the past few years, numerous bacterial strains have become resistant to selected drugs from various therapeutic groups. A potential tool in the fight against these strains is antimicrobial photodynamic therapy (APDT). APDT acts in a non-specific manner by generating reactive oxygen species and radicals, thereby inducing multidimensional intracellular effects. Importantly, the chance that bacteria will develop defense mechanisms against APDT is considered to be low. In our research, we performed the synthesis and physicochemical characterization of curcumin derivatives enriched with morpholine motifs. The obtained compounds were assessed regarding photostability, singlet oxygen generation, aggregation, and acute toxicity toward prokaryotic *Aliivibrio fischeri* cells in the Microtox^®^ test. The impact of the compounds on the survival of eukaryotic cells in the MTT assay was also tested (WM266-4, WM115—melanoma, MRC-5—lung fibroblasts, and PHDF—primary human dermal fibroblasts). Initial studies determining the photocytotoxicity, and thus the potential APDT usability, were conducted with the following microbial strains: *Candida albicans*, *Escherichia coli*, *Staphylococcus aureus*, *Streptococcus pneumoniae*, and *Pseudomonas aeruginosa*. It was noted that the exposure of bacteria to LED light at 470 nm (fluence: 30 J/cm^2^) in the presence of quaternized curcumin derivatives at the conc. of 10 µM led to a reduction in *Staphylococcus aureus* survival of over 5.4 log.

## 1. Introduction

In recent years, antimicrobial resistance (AMR) has become one of the most significant issues in healthcare and is considered to be one of the three most severe threats to human health by the World Health Organization [1,2,3]. More and more antibiotic-resistant strains, which were typically found in hospitals, such as carbapenem-resistant Enterobacterales (CRE), methicillin-resistant *Staphylococcus aureus* (MRSA), and vancomycin-resistant *Enterococci* (VRE), are currently observed in the environment [1,4]. In 2019, five million people died due to antimicrobial resistance, and by 2050, AMR will be responsible for 10 million deaths annually [5]. Each year, the number of antibiotic-resistant strains increases, primarily due to antibiotic overuse [2,5]. There are many ways and mechanisms by which bacteria gain antibiotic resistance. Environmental bacteria use evolution, horizontal gene transfer, and co-existence [2]. Several modes of action exist in bacteria and contribute to antibiotic resistance, including restricted drug uptake, drug export using the efflux pump, drug target mutations, and enzymatic drug inactivation [2,5]. The variety of ways and survival mechanisms that bacteria possess triggers scientists to look for other approaches to bacterial infection treatment.

One of the possible approaches is photodynamic antibacterial chemotherapy (PACT), also called antimicrobial photodynamic therapy (APDT) or antimicrobial photodynamic inactivation (aPDI/APDI) [6]. Photodynamic therapy (PDT) is a form of therapy where three factors—light, photosensitive compounds, and molecular oxygen—are used to generate reactive oxygen species (ROS) in order to treat diseases and cancers. Photodynamic antibacterial chemotherapy (PACT) is a form of photodynamic therapy targeted against microorganisms [7,8]. ROS are generated in the vicinity of the photosensitizer, which minimizes the side effects of PACT. After exposure to light, the photosensitizer’s energy increases, and the molecules enter sequentially into singlet and triplet states. Deactivation of the triplet state activates oxygen molecules, which results in the creation of ROS, including singlet oxygen. Singlet oxygen oxidizes surrounding biomolecules, leading to cell damage and, consequently, necrotic or apoptotic death [9,10]. One of the main advantages of PACT is its efficacy against AMR bacteria. The non-selective mode of action and achievement of various therapeutic targets and biomolecules prevent microorganisms from developing specific resistance mechanisms [8,9,10,11,12]. The efficacy of the therapy depends strongly on the photosensitizer’s structure. Considering this, numerous studies focus on obtaining molecules that may be used as drugs in PACT. They include natural and synthetic compounds, such as phenothiazines, xanthenes, riboflavins, curcuminoids, and porphyrinoids [9,10,13,14].

Curcumin, a natural compound isolated from the turmeric rhizome (*Curcuma longa* L.), has multidirectional biological effects. Both curcumin and its synthetic derivatives possess interesting prospects as potential antibiotics. Numerous studies have confirmed their antimicrobial activity [15,16,17]. The main obstacles associated with this group of compounds include low water solubility and low bioavailability. Related mechanisms of action include the removal of existing and prevention of new bacterial biofilm formations, cell membrane deterioration, and ROS generation after light exposure. In the literature, there are many examples of synergistic antimicrobial activity of curcumin in combination with antibiotics or antifungal drugs against various pathogens, including methicillin-resistant *Staphylococcus aureus*, as well as *Pseudomonas aeruginosa*, *Escherichia coli*, and *Candida albicans* [18,19,20,21]. Furthermore, a combination of curcumin derivatives’ anti-inflammatory and antimicrobial properties may be used in *Helicobacter pylori*-associated gastritis, peptic ulcers, and stomach adenocarcinoma [16]. As curcumin itself does not have a very pronounced effect, in our research, we decided to modify its chemical structure to enhance the activity by introducing groups that increase its antibacterial potential, such as quaternary ammonium moieties.

Quaternary ammonium compounds (QACs) are a large group of chemical compounds containing at least one nitrogen atom with a positive charge connected to at least one hydrophobic hydrocarbon scaffold. Additionally, many QACs have chloride or bromide anions in their structure and possess the ability to form micelles [22]. Due to their mode of action, they are used, among others, as disinfectants, surfactants, antiseptics, antimicrobials, and preservatives [22,23,24]. QACs’ antimicrobial effects are most important when considering public health and growing AMR. Their activity is related to their structure and based on cell wall and membrane disintegration. Positively charged nitrogen atoms interact with bacterial proteins and phospholipids in the membrane, which results in membrane destabilisation and cell lysis [22,25].

Considering curcumin’s potential, limited by low water solubility, and QAC’s antimicrobial activity, we obtained quaternary ammonium curcuminoids and studied them in terms of their physicochemical and biological properties. It was assumed that the introduction of the quaternary ammonium moieties to curcumin’s chemical structure ought to improve its bioactivity against bacteria and thus its prospective applications in antimicrobial photodynamic therapy.

## 2. Results

### 2.1. Chemical Synthesis and Characterization

In this study, two new curcumin derivatives (**2** and **3**) (Figure 1) based on the curcumin scaffold were synthesized by adapting a recently published methodology (Figure 2) [26,27]. BF_2_ protection was implemented for several reasons. Firstly, we wanted to avoid unexpected side reactions, mainly substitution reactions. Secondly, BF_2_ protection improves the chemical stability of the curcumin scaffold and thus often increases biological activity. Williamson ether synthesis in dry DMF with potassium carbonate acting as a base was used to modify para-hydroxybenzaldehyde and led to modified benzaldehyde with a 2-(morpholin-4-yl)ethoxy moiety with 41% yield [28,29]. Synthesis of **2** was carried out via an aldol condensation reaction in toluene with tributyl borate as a dehydrating agent and *N*-butylamine as a catalyst, with a high yield of 84% [30]. Product **2**, which precipitated after the condensation reaction from the reaction mixture, was treated with iodomethane in boiling chloroform overnight in order to obtain the quaternary derivative **3** [31]. The resulting orange solid **3** was obtained with a 51% yield after being purified by rinsing with a DCM:MeOH (10:1) mixture.

Curcumin derivatives **2** and **3** were characterized by 1D and 2D NMR (Figure 3, Figure 4, Figure 5 and Figure 6) and ESI-HRMS. The values of ^1^H and ^13^C chemical shifts and correlations between protons and carbons were carefully analyzed (Table 1 and Table 2).

### 2.2. Physicochemical Characteristics

#### 2.2.1. Spectral Properties

In the UV–vis spectra of the studied curcumin derivatives **2** and **3**, typical absorption bands in the 350–550 nm range were noted (Figure 7) [32,33,34]. The absorption maxima of **2** and its quaternized derivative **3** were located close to 500 nm in both DMSO and DMF (for **2**, λ_max_ 496 nm in DMF and 502 nm in DMSO; for **3**, λ_max_ 489 nm in DMF and 495 nm in DMSO), whereas for the unsubstituted curcumin, the absorption maximum was located at 430 nm in DMF and 436 nm in DMSO. This implies that the presence of the BF_2_ moiety induces a strong bathochromic shift, as reported before [35]. Further conversion of non-ionic 2-(morpholin-4-yl)ethoxy substituents present in **2** into quaternary ammonium cations present in **3** 2-(*N*-methylmorpholinium)ethoxy groups induced a slight 6 nm hypsochromic shift in both solvents. The strong absorption band originates mainly from π-π* electronic transition. To a lesser extent, such peaks come from the n-π* transition, although the importance of this crossing remains unclear [36,37].

#### 2.2.2. Photodecomposition Quantum Yields

The absorption spectra of the studied curcumin derivatives **2** and **3** were monitored during the irradiation and compared with curcumin and zinc(II) phthalocyanine (ZnPc) as references (Figure 8). Under visible light irradiation (λ ≥ 450 nm), curcumin derivatives **2** and **3** only slightly decomposed. The calculated quantum yields were at the level of 10^−4^. Compound **3** in DMF and curcumin showed lower stability than **2** (Table 3). Curcumin derivative **2** revealed the highest stability of the studied curcumins, and it was noted that it had a higher stability in DMF (Φ_P_ = 2.90 × 10^−5^) than in DMSO (Φ_P_ = 5.41 × 10^−5^). Considering the obtained results, the morpholinoethyl and BF_2_ moieties tend to stabilize the curcumin core and prevent its photodecomposition. Partial reduction in light absorption (expressed by ε values) has also been noted for other photosensitizers subjected to the quaternization when 2-(morpholin-4-yl)ethoxy substituents were transformed into 2-(*N*-methylmorpholinium)ethoxy ones [31].

#### 2.2.3. Singlet Oxygen Generation Quantum Yields

Singlet oxygen generation quantum yield assessment constitutes a crucial measurement for the prospective photosensitizers applied in photodynamic therapy. For the singlet oxygen generation study, the indirect method was chosen with two different singlet oxygen quenchers—1,3-diphenylisobenzofuran (DPBF) and tetrathiafulvalene (TTF). Due to the partial overlapping of the DPBF band with the studied compounds’ bands in absorption spectra, a singlet oxygen formation assessment was also performed with TTF as a chemical singlet oxygen scavenger. The values obtained for new curcuminoids **2** and **3** were referenced to curcumin and ZnPc.

Despite lower photostability and singlet oxygen generation in the experiment with DPBF than the reference ZnPc (Φ_P_ = 10.2 × 10^−6^ in DMF, 3.5 × 10^−6^ in DMSO [38]; Φ_Δ_ = 0.56 in DMF, 0.67 in DMSO [39]), the curcumin derivative **3** also revealed low values of singlet oxygen generation, reaching up to Φ_Δ_ = 0.1, when TTF was used (Table 4, Figure 9). The measured values were comparable with natural curcumin (estimated at 0.04–0.1) and tended to be consistent in various solvents. However, the measured quantum yield values in DMF were much higher than those previously reported for curcumin [40,41,42]. The singlet oxygen quantum yields for curcumin and its derivatives remained low in comparison to macrocyclic photosensitizers, such as ZnPc, probably due to the size of the π-delocalized electron system. Additionally, the singlet oxygen generation quantum yields may have been affected by the electron-donating and -withdrawing effects of the moieties attached to the curcumin scaffold, as reported before [34,35,37,43]. In the case of the studied curcuminoids **2** and **3**, the quantum yields of singlet oxygen generation firstly effectively decreased for **2**, while after changing its electronic characteristics by quaternization to **3***,* those values increased and were close to those of the reference curcumin. Interestingly, the obtained singlet oxygen quantum yields were similar for both singlet oxygen scavengers (DPBF and TTF) in DMSO but differed slightly when measured in DMF.

#### 2.2.4. Aggregation Studies

The aggregation properties of the studied compounds were evaluated using statistical analysis of the UV–vis absorption spectra during serial dilution. In DMF, compound **2** and the reference curcumin revealed strong aggregation, while for quaternized compound **3** the aggregation was negligible. This can be associated with the presence of a positive charge that prevents aggregation due to the repulsion of the same charges [37]. On the other hand, none of the compounds revealed strong aggregation in DMSO, considering the statistics and comparison of the intercept parameters with Student’s t-distribution values. According to the literature, DMSO can permeate into curcumin clusters and stabilize the non-aggregated molecules due to solvation [44].

### 2.3. Acute Toxicity Assessment—Microtox Assay

All studied compounds were subjected to a Microtox acute toxicity test. The test is based on bioluminescent *Aliivibrio fischeri* bacteria, which are very sensitive to the presence of toxic substances. Changes in the toxicity of the environment surrounding the bacterium influence its respiratory processes and translate into the emitted bioluminescence intensity. Therefore, in contact with a toxic substance at a specific concentration, a linear decrease in luminescence is observed [45]. Dimethylsulfoxide was used as a solubilizer of studied compounds in amounts not exceeding 1% *v*/*v*. To determine the effect of the solvent alone, 1% DMSO solution in water was also tested.

The results of the experiments are summarized in Figure 10. For compound **2**, a significant effect was observed only for the highest concentration tested (100 µM), with a 45% decrease in *A. fischeri* bioluminescence. For the lower concentrations, the effect was negligible and only slightly higher than for 1% DMSO solution in water; therefore, it can be regarded as non-toxic [46]. The quaternized derivative **3** exhibited a stronger effect at the same concentrations tested, which is probably linked to the toxicity of the cationic nature of quaternary ammonium salts [47]. In this case, the effect was observed at 100 µM (99%) and 10 µM (53% after 5 min exposure). Based on the results obtained, compound **3** was found to be more toxic than **2**. Additionally, in order to compare the activity of compounds **2** and **3**, the Microtox test was performed for reference compounds such as curcumin and the curcumin–BF_2_ complex. The results indicate that the curcumin–BF_2_ complex is toxic at every concentration tested, whereas curcumin reveals ecotoxicity only at concentrations of 100 µM and 10 µM. Potentially, this sudden change in the bioactivity of curcumin from 99% (10 µM) to 0% (1 µM) is related to the fluorescent properties of curcumin and the potential distortion of the results [36]. Nevertheless, the acute toxicity of compound **2** is lower than that of curcumin, while the results obtained for compound **3** are comparable to those for curcumin.

### 2.4. In Vitro Photodynamic Antimicrobial Activity

In the first experiment, the main goal was to determine whether the tested compounds were antimicrobially active with or without light. Two compounds (**2** and **3**) were tested in the visible range, and a 470 nm LED lamp was applied (Figure 11). Due to poor solubility and low absorption, curcumin was not included in the tests. A screening study was carried out on three strains: Gram-positive bacteria—*Staphylococcus aureus* NCTC 4163, Gram-negative bacteria—*Escherichia coli* ATCC 25922, and the yeast *Candida albicans* ATCC 10231. In the study, the irradiation lasted 30 min, and the lamp was placed 10 mm above the microtiter plate containing the bacterial suspensions and tested compounds at concentrations of 10 μM. The preincubation time (PIT) lasted 20 min, and the LED lamp emitted visible light (λ_max_ 470 nm) with an irradiance of 16.67 mW/cm^2^, which corresponds to a light dose (fluence) of 30.01 J/cm^2^ after 30 min of exposure. The light exposure conditions (lamp distance, lamp power, and lamp type) were the same in all subsequent experiments. The light dose depended on the irradiation time. The light used alone (without the addition of compounds) did not show any activity against any microorganism tested. According to the results, none of the tested compounds revealed antifungal activity against *Candida albicans* (Table 5). Curcumin derivative **2** showed a 2.13 log reduction in bacterial cell viability against *S. aureus* bacteria and no activity against *E. coli*. Compound **3** presented activity against both G+ (*S. aureus*) and G− (*E. coli*) bacteria. The light-induced antimicrobial activity against *S. aureus* and *E.coli* exceeded log reductions of 5.54 and 4.46, respectively. At these concentrations, not even a single colony was observed on the plates, as complete eradication occurred.

The second experiment was designed to determine the time and concentration required to achieve the biological effect observed in the first experiment and to extend the study to other strains (Figure 12). Apart from *S. aureus* and *E. coli*, the Gram-positive *Streptococcus pyogenes* ATCC 19615, which is the etiologic agent of tonsillitis, and the Gram-negative *Pseudomonas aeruginosa* NCTC 6749, which is a major threat to hospitalized and immunocompromised patients due to its high antibiotic resistance, were used. The observed effect was strictly concentration-dependent (Table 6). According to the results, Gram-positive bacteria appear to be the most susceptible to the applied photocytotoxicity study. Complete eradication of *S. aureus* with compound **3** at a concentration of 10 µM occurred after 15 min of irradiation, whereas no biological effect was observed after 25 min of irradiation at a concentration of 1 µM. For *S. pyogenes*, the results were even more pronounced, as the complete eradication under the influence of compound **3** at a concentration of 10 µM occurred after only 5 min of irradiation, while at a concentration of 1 µM, complete eradication occurred after 15 min of irradiation. Interesting results were also obtained for Gram-negative bacteria. The highest log reduction obtained for *E. coli* bacteria was observed for **3** after 25 min of exposure to light at the concentration of 10 µM, whereas no biological activity was noted at the lower concentration of 1 µM. A high level of eradication of *P. aeruginosa* in the presence of compound **3** (10 µM) was observed after only 5 min of irradiation (log reduction: 4.29). However, complete eradication did not occur until 25 min of irradiation. Reductions in microbial survival were still observed with a decrease in compound concentration, but complete eradication was not achieved within the irradiation time.

As the fluence [J/cm^2^] depends on the time of exposure [s] and irradiance [mW/cm^2^], it was calculated for the time points at which the tested microorganisms were eradicated to a degree of 99.9% (log reduction > 3). The results are shown in Table 7.

In the third experiment, we decided to disprove that the photodegradation products of compound **3** were responsible for the biological effects observed in the previous experiments (Table 8). To verify this, we performed a test using *S. aureus* and *E. coli* strains with compound **3** at a concentration of 10 µM (Figure 13). In this case, instead of using a solution containing the studied compound, a solution of the studied compound previously exposed to 30 min irradiation under light and time conditions consistent with the previous experiments was added to the bacterial suspensions. In the control, a saline solution (0.9% NaCl) was added to the bacterial suspension instead of the studied compound’s solution. The results clearly confirmed that the observed effect was not caused by the photodegradation products of compound **3**.

To conclude, the microbiological experiments indicated that compound **3** at the concentration of 10 µM eradicated *S. aureus* completely after 15 min, *S. pyogenes* after 5 min, and *P. aeruginosa* after 25 min of irradiation. It is worth noting that compound **3** also revealed the activity at the concentration of 1 µM, allowing for complete eradication of *S. pyogenes* after 15 min of irradiation. It is also essential that the biological effects of the photodegradation products of compound **3** were disproved.

### 2.5. Cell Viability Assessment

To determine the toxicity of the tested compounds against normal and tumor cells, a cell viability assay was performed using MTT dye. The following cell lines were used: PHDF—primary human dermal fibroblasts, MRC-5—lung fibroblasts, WM266-4—melanoma, and WM115—melanoma. Whenever possible, the IC_50_ value was calculated for each cell line. The IC_50_ results are reported in Table 9, while Figure 14 presents graphs illustrating the dependence of cell viability on the concentration of the tested compounds.

The responses of cancer and normal cells to curcumin were similar, and all cells revealed similar sensitivity to curcumin, with IC_50_ values at the micromolar level. Compounds **2** and **3** presented slightly higher activity towards normal cells than cancer cells. Based on this, it can be concluded that compounds **2** and **3** are not good candidates for anticancer drugs. Additionally, compound **3** did not reveal a toxic effect on healthy skin and lung fibroblast cells. The toxicity of the tested curcumin derivative **3** against eukaryotic lines was also lower than that of curcumin.

The results can be compared with the research of other scientific teams. In experiments performed by Lazewski et al. [32], curcumin was tested on MRC-5 fibroblasts, and the IC_50_ values were found to be 70.35 ± 12.45 µM and 45.33 ± 3.18 µM after 24 h and 48 h, respectively. In our study, the values were slightly lower. However, the basic relationship did not change because, after 48 h, the value of curcumin concentration necessary for a reduction in cell viability by 50% was lower due to a longer exposure time to the agent and a greater chance of reaching intracellular structures. In another study by Siwak et al. [49], a similar experiment with curcumin was performed on the WM266-4 melanoma cell line. The scientists found that the IC_50_ value of curcumin was about 7 µM after 96 h of incubation. Considering the observed trend (IC_50_ concentration decreased with increasing incubation time), the values of 45 µM after 24 h and 21 µM after 48 h presented in the publication were convergent.

## 3. Discussion

Curcumin derivatives with quaternary morpholine moieties **3** were synthesized and subjected to detailed physicochemical and microbiological characterization. The physicochemical study of curcumin derivatives **2** and **3** concerned photostability and photochemistry, as well as aggregation assessment. In addition, antimicrobial and acute toxicity studies of the curcumin derivatives were performed with the Microtox test.

The obtained biological activity data can be viewed in a broader context when trying to assess the value of the research conducted. It seems that the use of a morpholine substituent in the structure was of great importance for the obtained physicochemical and biological parameters of the new compound, **3**. Several studies which are available in the literature indicated the validity of using the morpholine moiety in the structure of compounds of photochemical potential for PDT [28,50]. Promising data were also presented for molecules equipped with quaternary morpholine moieties, which could be potentially applied in PACT [13,31,51]. So far, effective light-induced antimicrobial activity has been proven in the context of quaternary iodide salt of magnesium(II) phthalocyanine with *N*-methyl morpholiniumethoxy substituents [31]. Achieving full eradication at a level of more than 5 log reduction for *S. aureus* (including MSSA and MRSA), *E. coli*, and *P. aeruginosa* was possible using an LED lamp (λ_max_ = 735 nm) and a light dose of about 1.8 J/cm^2^ (3 mW/cm^2^, 10 min). In the study, the photosensitizer concentration was 100 µM (1 × 10^−4^ M). Further in-depth studies on the same compound were carried out, with more emphasis on the eradication of *P. aeruginosa* [13]. Twenty-nine clinical strains isolated from patients’ lower respiratory tracts or chronic wounds were used for the study. A reduction of more than 4 log was achieved with light conditions similar to those mentioned above. Researchers from the team of Prof. T. Nyokong studied similar macrocyclic compounds containing quaternized morpholine in their structure [51]. In the study, derivatives of phthalocyanines with quaternized morpholine groups were obtained and studied against, among other strains, *E. coli*, *S. aureus* (including MRSA), and *C. albicans*. The photosensitizer concentration was optimized with a dose of 5 µM, which was used in further studies. The laser light dose (λ_max_ 680/690 nm) necessary to eradicate the mentioned microorganisms entirely was 108 J/cm^2^ (30 mW/cm^2^, 1 h). The exposure times and light fluence used to induce the biological effect varied, but the research conducted in this work does not differ significantly in terms of these parameters from the publications of other authors. When it comes to testing curcumin derivatives for APDT, one of the most interesting compound is SACUR-3, which was analyzed in Plaetzer’s lab [52,53]. This cationic curcumin derivative at a dose of 50 µM revealed an inactivating effect on microorganisms in an ex vivo porcine skin model after irradiation with LED light with a maximum of 435 nm at a dose of 33.8 J/cm^2^. Unfortunately, the research cannot be directly compared to ours, due to the fact that our research was not conducted on porcine skin. It is interesting to note that the study indicated the potential utility of the SACUR-3 compound in the photodynamic decontamination of food. To better present the context of the conducted research, a comparison of selected scientific reports involving similar strains of microorganisms is presented below, in Table 10. The APDT biological activities of popular dyes such as methylene blue, toluidine blue, and rose bengal, as well as exemplary macrocyclic and BODIPY dyes, are included.

The most active compound, **3**, was analyzed in terms of photocytotoxicity on five strains of microorganisms: two Gram-positive strains (*Staphylococcus aureus* and *Streptococcus pyogenes*), two Gram-negative strains (*Escherichia coli* and *Pseudomonas aeruginosa*), and a fungus (*Candida albicans*). The study allowed us to determine the photocytotoxic potential of this molecule against microorganisms with different pathogenicities. In 2017, the World Health Organization (WHO) published a list of multidrug-resistant bacterial strains for which drugs are urgently sought [62]. This list includes, among others, the strains used in our research, such as *Pseudomonas aeruginosa*, *Escherichia coli*, and *Staphylococcus aureus*. Compound **3** at the concentration of 10 µM allowed complete eradication of *S. aureus* after 15 min, *S. pyogenes* after 5 min, and *P. aeruginosa* after 25 min of irradiation. It is worth noting that compound **3** also revealed activity at the concentration of 1 µM, allowing for complete eradication of *S. pyogenes* after 15 min of irradiation. It is also important that the biological effects of the photodegradation products of compound **3** were disproved.

It was found that the new curcumin derivative could be considered as a potential photosensitizer in antimicrobial photodynamic therapy only against bacteria, as activity towards fungi was not noted. The results of our research confirm that, among the microorganisms used in our study, the fungus *Candida albicans* is the most resistant to photodynamic therapy and that a reduction in its viability was not possible under the tested conditions. This is interesting because in different experiments conducted so far by other research groups, even unmodified curcumin allowed such an effect to be achieved [63,64,65]. It is worth emphasizing, however, that an LED lamp with a slightly different maximum (λ_max_ 455 nm) was used to irradiate the unmodified curcumin, and the effect was obtained at a concentration of 20 µM and a fluence of 5.28 J/cm^2^ [64]. We suspect that the lack of activity of our compounds may have been due to insufficient pre-illumination time (PIT), but available data in the literature allow us to exclude this hypothesis in relation to studies on the planktonic *C. albicans* [66]. In vivo studies were also carried out on a mouse model of oral candidiasis, which showed that photodynamic therapy with curcumin (80 µM curcumin, 37.5 J/cm^2^) resulted in a reduction in the survival of *C. albicans* by 4 log [65]. Particularly noteworthy is the very good effect of compound **3** against *P. aeruginosa*, a species that is naturally very resistant to antimicrobials, as it has natural and acquired resistance to various antibiotics and is more resistant than other microorganisms to disinfectants and antiseptics [67,68].

Another aspect of our research concerns the application of light at a proper wavelength. Research shows that UV light (<400 nm) has a negative impact on health and skin. It can cause genetic mutations and consequently increase the risk of cancer. However, this depends on the light dose. Studies indicate the toxicity of light with a λ_max_ of 453 nm and a fluence >500 J/cm^2^ [69]. This means that the therapy we propose should not have toxic effects on healthy cells, as we used much lower doses of light (16–100 times lower with a fluence rate of 5–30 J/cm^2^). Research on blue light with a maximum of 470 nm and a fluence of 5 J/cm^2^ showed that it does not impair wound healing in vitro and can even stimulate the tissue regeneration process [70]. Typically, the light dose (fluence and radiant exposure) during photodynamic therapy directed against cancer is in the range of 20–200 J/cm^2^ [71], whereas under photodynamic therapy directed against microorganisms, it is about 5–100 J/cm^2^ [6,12,72]. In our case, the fluence necessary to observe the desired biological effects is in the range of 5–30 J/cm^2^, which does not differ from the values used in other publications.

An essential aspect taken up in our studies concerns the assessment of the toxicity of the tested compound on healthy tissues. This type of assessment cannot be omitted because many compounds that are potentially active against microbes also exhibit high cytotoxicity in healthy tissues. Our research fortunately revealed that antibacterial compound **3** has no toxic effect on normal human cells (PHDF—primary human dermal fibroblasts and MRC-5—lung fibroblasts) or skin cancer cells (WM266-4—melanoma and WM115—melanoma). The IC_50_ values for the mentioned lines were above 50 µM and exceeded the concentration of compound **3** used for antimicrobial testing (1–10 µM) by 5 to 50 times. Ecotoxicity testing using the Microtox method against bioluminescent Gram-negative bacteria *Aliivibrio fischeri* showed that compound **3** could be ecotoxic at concentrations of 10 µM and 100 µM. This result is consistent with the data obtained during microbiological tests, as the Microtox test is also based on the bacterium *Aliivibrio fischeri*.

## 4. Materials and Methods

### 4.1. Chemistry

High-grade chemical reagents and solvents were obtained from commercial vendors (Fluorochem, Hadfield, UK; Tokyo Chemical Industry—TCI, Tokyo, Japan; Acros Organics, Geel, Belgium; Sigma Aldrich, St. Louis, MO, USA) and were used without additional purification. Curcumin was purchased from TCI and curcumin-BF_2_ was synthesized according to a previously published procedure [32]. *N*,*N*-Dimethylformamide (DMF) and dimethylsulfoxide (DMSO) were purchased from Fisher Scientific (Waltham, MA, USA); 1,3-diphenylisobenzofuran (DPBF) and tetrathiafulvalene (TTF) were purchased from Sigma Aldrich. Thin-layer chromatography (TLC) was performed on precoated TLC plates (SiliaPlate TLC, thickness 200 µm, F-254; SiliCycle Inc., Quebec City, QC, Canada) and visualized under a UV lamp (λ = 254 and 365 nm). Chromatographic preparative separations were accomplished in glass columns filled with silica gel (SiliaFlash P60, particle size 40–63 µm; SiliCycle Inc., Quebec City, QC, Canada). Melting points were determined in open glass capillaries using a Stuart SMP10 apparatus (Bibby Sterilin Ltd., Stone/Staffordshire, UK) and were uncorrected. Nuclear magnetic resonance (NMR) spectra (1D: ^1^H and ^13^C; 2D: ^1^H-^1^H COSY, ^1^H-^13^C HMBC, and ^1^H-^13^C HSQC) were recorded on an AvanceCore 400 MHz spectrometer (Bruker, Fällanden, Switzerland). High-resolution mass spectra (HR-MS) were recorded on a Bruker Impact HD apparatus (Bruker Daltonics, Billerica, MA, USA) operating in electrospray mode (ESI) with positive ionization.

#### 4.1.1. Synthesis of Modified Aldehyde—Stage I

The synthesis of modified aldehyde containing a morpholinethoxy group was performed according to the literature and our previous experiments [28,29] via modified Williamson ether synthesis. Intermediate para-hydroxybenzoic aldehyde functionalized with a morpholinoethoxy group was obtained as follows. First, 2.500 g of 4-hydroxybenzoic aldehyde (20.5 mmol, 1.0 equiv.), 3.679 g of 4-(2-chloroethyl)morpholine hydrochloride (24.6 mmol, 1.2 equiv.), and 16.993 g of potassium carbonate (123.0 mmol, 6 equiv.) were transferred to a dry round-bottom flask. Then, 30 mL of *N*,*N*-dimethylformamide (DMF) was added, and the reaction was stirred for 24 h under nitrogen at 70 °C. The resulting mixture was cooled to ambient temperature and filtered to remove K_2_CO_3_. The filtrate was evaporated to give a yellowish oil, which was then purified by column chromatography (DCM: MeOH 25:1), resulting in a colorless oil (1.971 g, 41% yield). The product was stored in a fridge (2–8 °C) to avoid decomposition.

#### 4.1.2. Synthesis of Acetylacetone–BF_2_—Stage II

The synthesis of acetylacetone blocked with a BF_2_ group was accomplished according to the procedures reported in the literature [26,27,30]. That is, 5.14 mL of acetylacetone (5.00 g, 50.0 mmol, 1 equiv.) was dissolved in 100 mL of dichloromethane (DCM) in a dry round-bottom flask. After that, 9.3 mL (10.66 g, 75.0 mmol, 1.5 equiv.) of boron trifluoride diethyl etherate was added, and the reaction mixture was stirred for 24 h at 40 °C under nitrogen. Afterwards, the reaction was cooled to ambient temperature and quenched by the addition of water (100 mL). The organic layer was extracted several times with H_2_O until the pH of the aqueous layer became neutral. Further, the organic phase was dried with anhydrous sodium sulfate and evaporated under reduced pressure. The obtained product was an oil that finally crystallized, giving long transparent needles with a slightly brownish tint (5.304 g, 72% yield).

#### 4.1.3. Synthesis of **2** via Aldol Condensation—Stage III

The synthesis of curcuminoid **2** followed the aldol condensation mechanism and was performed in accordance with the procedure described earlier in the literature [30]. Initially, 500 mg of acetylacetone–BF_2_ (3.36 mmol, 1.0 eq.) and 1.580 g of morpholinethoxy-functionalized aldehyde (6.72 mmol, 2 eq.) were transferred to a dry round-bottom flask. Subsequently, 20 mL of toluene, 1.812 mL of tributyl borate (1.545 g, 6.72 mmol, 2.0 eq.), and 66 µL of *N*-butylamine (49 mg, 0.67 mmol, 0.2 eq.) were added consecutively. The reaction was continued for 24 h under inert gas at 70 °C with stirring. Next, the reaction mixture was cooled to ambient temperature, and the resulting reddish precipitate was filtered. The solid was washed several times with toluene and dried in the air, leading to **2** (1.632 g, 84% yield).

Compound **2** Yield: 84%; ESI MS found: 583.2783 *m*/*z* [M + H]^+^, expected for C_31_H_37_BF_2_N_2_O_6_: 583.2785 *m*/*z* [M + H]^+^; R*_f_* 0.52 (DCM:MeOH 10:1); mp 170–173 °C; UV-vis λ_max_ (ACN) nm (logε): 487 (4.88); ^1^H NMR (400 MHz, DMSO-*d*_6_) δ 7.98 (d, *J* = 15.6 Hz, 2H), 7.84 (d, *J* = 8.9 Hz, 4H), 7.08 (dd, *J* = 12.3, 3.4 Hz, 6H), 6.50 (s, 1H), 4.19 (t, *J* = 5.7 Hz, 4H), 3.62–3.53 (m, 8H), 2.71 (t, *J* = 5.7 Hz, 4H), 2.49–2.41 (m, 8H); ^13^C NMR (101 MHz, DMSO-*d*_6_) δ 179.6, 162.3, 146.8, 132.2, 127.4, 119.2, 115.8, 102.1, 66.6, 66.2, 57.3, 54.1.

#### 4.1.4. Synthesis of Compound 3—Stage IV

Compound **2** (200 mg, 0.34 mmol, 1.0 eq.) and 10 mL of chloroform were added to a dry round-bottom flask. Next, 340 µL of iodomethane (775.9 mg, 5.47 mmol, 16.0 eq.) was added to the reaction mixture, and the mixing was continued under reflux for 24 h. After that time, a bright orange precipitate was filtered and washed extensively with DCM:MeOH 10:1 to give pure product **3** (151 mg, 51% yield).

Compound **3** Yield: 51%; ESI MS recorded: 597.2949 *m*/*z* [M-2I-CH_3_]^+^, expected for C_33_H_43_BF_2_I_2_N_2_O_6_: 597.2942 *m*/*z* [M-2I-CH_3_]^+^; mp 253–255 °C; UV-vis λ_max_ (ACN) 476 nm; ^1^H NMR (400 MHz, DMSO-*d*_6_) δ 8.02 (d, *J* = 15.6 Hz, 2H), 7.91 (d, *J* = 8.9 Hz, 4H), 7.16–7.11 (m, 6H), 6.54 (s, 1H), 4.64–4.57 (m, 4H), 3.98 (t, *J* = 4.7 Hz, 12H), 3.62–3.50 (m, 8H), 3.28 (s, 6H).; ^13^C NMR (101 MHz, DMSO-*d*_6_) δ 179.4, 160.5, 146.3, 131.7, 127.7, 119.2, 115.6, 101.8, 62.3, 61.3, 59.8, 59.8, 47.2.

### 4.2. Physicochemical Studies

#### 4.2.1. Photodecomposition Quantum Yields

The photostability measurements were conducted in DMSO and DMF at ambient temperature under aerobic conditions according to the previously described method [73,74,75,76]. A 150 W xenon lamp (Optel, Opole, Poland) was utilized as a light source. In order to separate the visible light (>450 nm), the HCC-16 cutting filter was used. The spectral changes under the irradiation were recorded with an OceanOptics Flame spectrophotometer equipped with a DT-MINI-2-GS light source (OceanOptics, Dunedin, FL, USA). The photodegradation quantum yields were then calculated using the previously presented equation [39,73,74,76].

#### 4.2.2. Singlet Oxygen Generation Quantum Yields under Light Irradiation

Singlet oxygen quantum yield measurements for **2**, **3**, and curcumin were assessed in dimethyl sulfoxide (DMSO; Fisher, Longborough, UK) and *N*,*N*-dimethylformamide (DMF; Fisher) at ambient temperature according to the comparative method protocols described previously [39,73,74,75]. The method involves using chemical quenchers of singlet oxygen, 1,3-diphenylisobenzofuran (DPBF; Merck, Steinheim, Germany), tetrathiafulvalene (TTF; Aldrich, Steinheim, Germany), and an unsubstituted zinc(II) phthalocyanine (ZnPc; Aldrich) as a reference compound. The mixtures of curcumins and DPBF or TTF were irradiated with light adjusted to the maximum absorbance wavelength through the M250 monochromator (Optel, Opole, Poland). The changes in spectra under the irradiation were recorded with an OceanOptics Flame spectrophotometer equipped with a DT-MINI-2-GS light source. The singlet oxygen quantum yields were calculated using the previously presented equation [39,73,74,75].

#### 4.2.3. Aggregation Studies

The aggregation studies were performed at ambient temperature. Known amounts of **2**, **3**, and curcumin as a reference in volumetric flasks were dissolved in DMSO and DMF and left to stir overnight. Then, the solutions were diluted appropriately to achieve concentrations in the range between 1.5 × 10^−6^ mol/dm^3^ and 5.5 × 10^−5^ mol/dm^3^. The samples were then moved into quartz cuvettes (l = 10 mm, Hellma, Müllheim, Germany), and their spectra were recorded using a Shimadzu U-1900 spectrophotometer (Shimadzu, Kyoto, Japan). The recorded spectra and numerical data were then used for the determination of the dependence between the maximum peak absorption and the sample concentration according to the A = f(c) function, used further for statistical verification of the Lambert–Beer law.

### 4.3. Biological Activity

#### 4.3.1. Microtox Assay

The acute toxicity of the tested curcumionoids was investigated on a Modern Water Microtox Model 500 apparatus equipped with Modern Water MicrotoxOmni 4.2 software, according to the 81.9% screening test procedure provided by the manufacturer (Modern Water, London, UK). All the reagents and solutions (Microtox Acute Reagent, Microtox Diluent, and Microtox Osmotic Adjusting Solution) were purchased from Tigret (Warsaw, Poland). Solutions of the tested compounds were prepared using DMSO and deionized water. The concentration of DMSO did not exceed 1% *v*/*v*. Each compound was tested at three final concentrations (1 µM, 10 µM, and 100 µM) and at two time points: 5 min and 15 min. The change in the bacterial bioluminescence was monitored after the addition of the sample solution and compared to the negative control.

#### 4.3.2. In Vitro Photodynamic Antimicrobial Activity

Organisms and growth conditions

The microorganisms used in the study were (*i*) standard strains of bacteria: *Staphylococcus aureus* NCTC 4163, *Escherichia coli* ATCC 25922, *Pseudomonas aeruginosa* NCTC 6749, and *Streptococcus pneumoniae* ATCC 19615, and (*ii*) fungi: *Candida albicans* ATCC 10231. The standard strains were obtained from the National Collection of Type Cultures (NCTC, Salisbury, UK) and the American Type Culture Collection (ATCC, Manassas, VA, USA). Bacterial and yeast strains were stored in Microbank cryogenic vials (Pro-Lab Diagnostics, Richmond Hill, ON, Canada) at −70 °C ± 10 °C. The bacteria were cultured aerobically in brain–heart infusion (BHI) broth (OXOID, Basingstoke, UK) at 36 °C ± 1 °C for 20 h. *C. albicans* cultures were grown in Sabouraud dextrose broth (SDB, Merck, Steinheim, Germany) at 36 °C for 20 h.

In vitro photodynamic inactivation of planktonic cells

The organisms were harvested by centrifugation (3000× *g* for 15 min) and resuspended in 1.5 mL of 10 mM PBS, pH, 7.0 (Sigma-Aldrich, St. Louis, MO, USA). The cells were then diluted 1/100 in PBS to a final concentration of about 10^7^ colony-forming units (CFUs)/mL. Aliquots (100 μL) of a standardized microbial suspension were placed in the wells of microtiter plates, and an equal volume of a solution of the tested compound was added to give final concentrations of 1 × 10^−5^ (10 µM) and 1 × 10^−6^ M (1 µM) (groups P+). Negative control wells contained the bacterial or yeast suspension and PBS without the photosensitizer (groups P−). All the samples were incubated in the dark for 20 min and irradiated by the LED panel at a distance of 10 mm for a specific period, depending on the experiment (groups L+). To evaluate the antimicrobial activity of the tested compounds, duplicate experimental samples were prepared but not irradiated (groups L−). Viable microorganisms were measured by counting the number of colony-forming units (CFUs) after appropriate serial dilution and culture on tryptic soy agar plates (TSA, OXOID, Basingstoke, UK) or Columbia Blood Agar with Sheep Blood (CAS, OXOID, Basingstoke, UK) for bacteria and Sabouraud dextrose agar (SDA, Merck, Steinheim, Germany) for fungi, following incubation at 36 °C ± 1 °C for 24–72 h. The number of surviving organisms and the log_10_ reduction in microbial cells for each sample were calculated using a formula available in the literature [77].

Light parameters

Lamp type—LED λ_max_~470 nm (LIU470A LED Array Light Source; Thorlabs Inc., Bergkirchen, Germany); irradiance—16.67 mW/cm^2^ (measured with PM16-130 Power Meter, Thorlabs Inc., Bergkirchen, Germany, and Optel Opole Sp. z o.o., Opole, Poland).

#### 4.3.3. MTT Assay and Cell Culture

In order to analyze the cytotoxic effect of curcumin and compounds **2** and **3**, different cell lines were used: PHDF—primary human dermal fibroblasts, MRC-5—lung fibroblasts, WM266-4—melanoma (low-melanotic), and WM115—melanoma (high-melanotic). All cell lines were seeded onto 96-well flat-bottom plates at a concentration of 15,000 cells/well. Curcumin (**1**) and compounds **2** and **3** were dissolved in the completed culture medium at the final concentrations of 5 µM, 10 µM, 20 µM, 30 µM, 40 µM, and 50 µM. The final volume of medium in each well was 200 µL. Cells which were not exposed to the tested compounds constituted a control group. After 24 h, each of the reagents was added to the cells. Control cells were not treated with synthesized reagents; they were incubated with a completed cell culture medium. The incubation of cells with **1**, **2**, or **3** lasted 24 and 48 h. Then, the medium was discarded, and the new medium containing MTT (3-(4,5-dimethylthiazol-2-yl)-2,5-diphenyltetrazolium bromide) (Affymetrix, Cleveland, OH, USA) at a final concentration of 0.5 mg/mL was added to the cell culture. Cells were incubated for 2.5 h in cell culture conditions. Next, the medium was removed, and 100 µL of DMSO (Thermo Scientific, Waltham, MA, USA) was added to dissolve the formed formazan crystals. The absorbance was read with a Multiskan plate reader at 570 nm and the background at 655 nm (Thermo Scientific, Waltham, MA, USA). The experiment was performed in triplicate.

PHDF cells were extracted from skin pieces collected following mastectomy procedures at the Greater Poland Cancer Centre in Poznań, Poland. Each time, informed patient agreement was acquired for the “The Cancer Genome Atlas (TCGA)” project to utilize biological material for scientific research purposes. The researchers did not know the identity of the patients [78]. The MRC-5, WM266-4, and WM115 cell lines were obtained from the American Type Culture Collection (ATCC). For culturing of the PHDF, WM266-4, and WM115 cell lines, the medium contained Dulbecco′s Modified Eagle′s Medium (DMEM) (Biowest, Nuaillé, France), 10% of fetal bovine serum (Biowest, Nuaillé, France), and 1% of penicillin/streptomycin agents (Merck Millipore, Darmstadt, Germany). Cells were incubated in a humidified incubator containing 5% CO_2_. The MRC-5 cell line was cultured similarly to the method described in our previous article [77]. Briefly, cells were incubated at 37 °C in an atmosphere enriched with 5% CO_2_. The basic culture medium was Dulbecco’s Modified Eagle’s Medium (DMEM) (Biowest, Nuaillé, France) supplemented with 10% fetal bovine serum (FBS) (Biowest, Nuaillé, France) with the addition of 2 mM L-glutamine and 5% non-essential amino acid solution.

## 5. Conclusions

In recent years, we have witnessed and experienced the growing problem of antibiotic resistance and, as a result, the decreased sensitivity of microorganisms to available chemotherapeutics. In light of these reports, developing new therapies and agents is necessary. The presented studies contributed to this trend of research in the field of new photosensitizers for PACT therapy. New curcumin derivatives were synthesized, and their properties were compared with a naturally occurring reference compound—curcumin of well-documented activity. Studies were conducted both on prokaryotic and eukaryotic cells, and the environmental impact of the tested compounds was assessed. The results indicate that compound **3** has interesting antibacterial potential and therefore could be considered for further in-depth biological and physicochemical studies. A detailed study would allow us to understand its mechanism of action and also check whether the observed activity is directed against other microbial strains. Subsequently, it would be worth conducting research on normal and cancer cell lines using similar exposure conditions in both in vitro 2D and 3D models, as well as in vivo studies. It would also be possible to additionally characterize the safety profile of the light used, as the state of knowledge on this subject still requires further exploration. Additionally, it is also worth checking the possible synergism of antibacterial action with other active pharmaceutical ingredients as well as bioactivity against bacterial biofilms. Such extended research would allow us to indicate the potential of compound **3** for the treatment of bacterial infections, especially ones localized in wounds that are difficult to heal and other areas that can be reached with an appropriate lamp. Considering the low toxicity of curcumin derivative **3**, it may also be interesting to evaluate the chances of its use as a surface disinfectant.

## Figures and Tables

**Figure 1 molecules-29-04536-f001:**
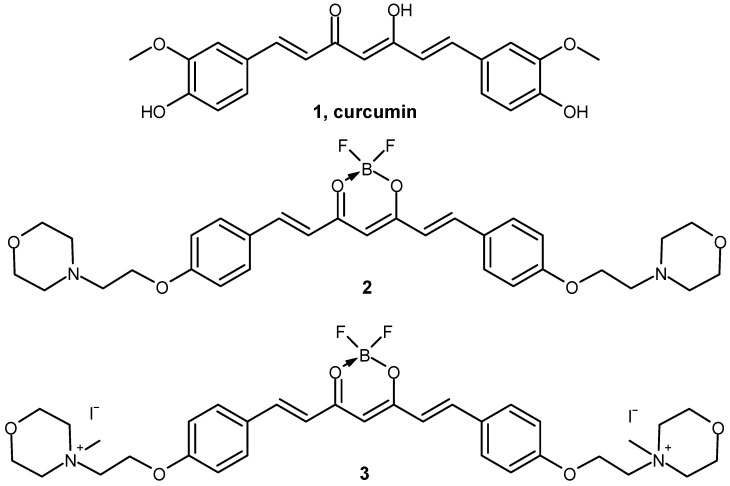
Chemical structures of compounds **1**–**3** involved in the study.

**Figure 2 molecules-29-04536-f002:**
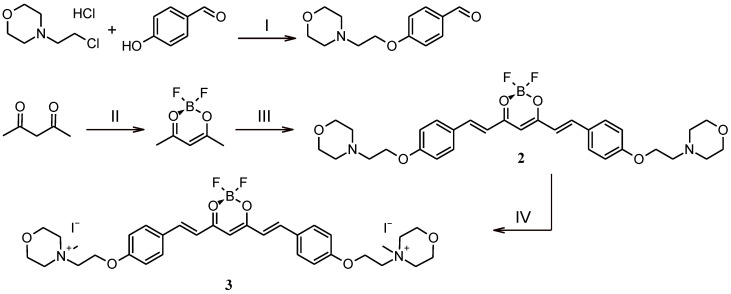
Synthetic pathways leading to curcumin derivatives **2** and **3**. Reagents and conditions: (I) DMF, K_2_CO_3_, 65 °C, overnight; (II) CH_2_Cl_2_, BF_3_·Et_2_O, 40 °C, overnight; (III) toluene, 4-(2-[morpholin-4-yl]ethoxy)benzaldehyde, tributyl borate, *N*–butylamine, 70 °C, overnight; (IV) CHCl_3_, CH_3_I, 70 °C, overnight.

**Figure 3 molecules-29-04536-f003:**
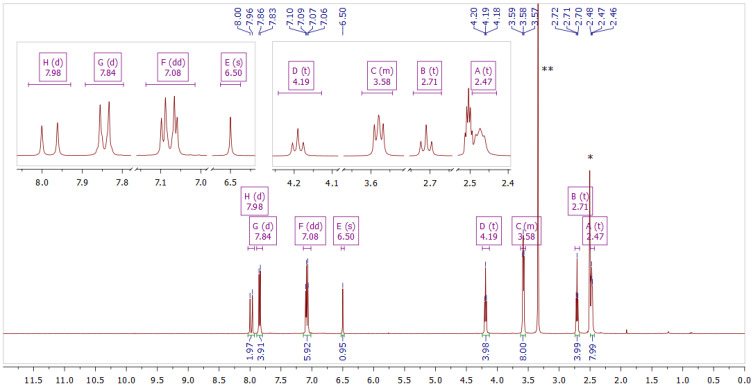
NMR spectrum of compound **2** in DMSO–*d*_6_. The inset shows fragments of the spectrum. * Solvent residual peak from DMSO (2.50 ppm). ** Solvent peak from H_2_O (3.33 ppm).

**Figure 4 molecules-29-04536-f004:**
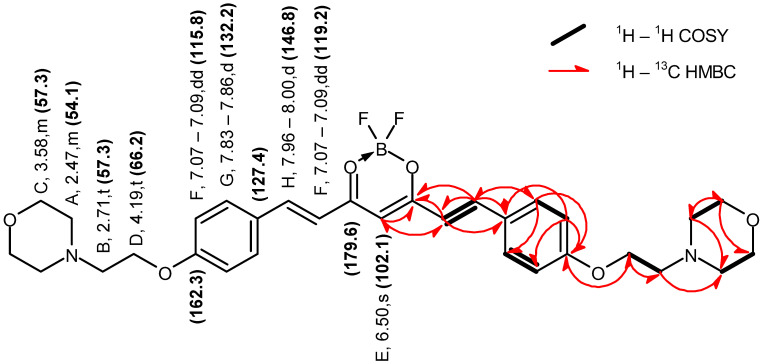
Graphical representation of the key NMR correlations (^1^H–^1^H COSY and ^1^H–^13^C HMBC) of compound **2** in DMSO–*d*_6_. The values of chemical shifts are expressed in ppm.

**Figure 5 molecules-29-04536-f005:**
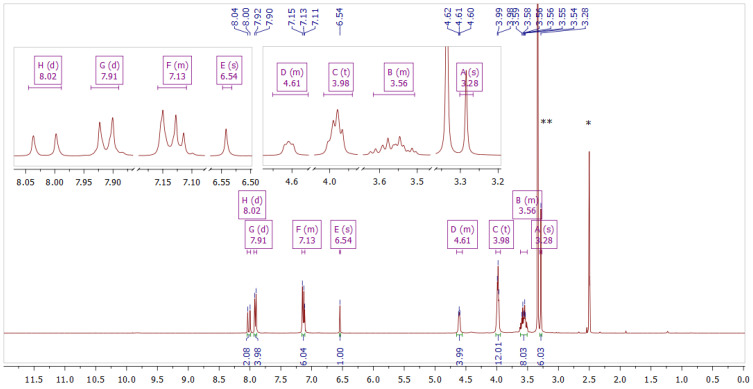
NMR spectrum of compound **3** in DMSO–*d*_6_. The inset shows fragments of the spectrum. * Solvent residual peak from DMSO (2.50 ppm). ** Solvent peak from H_2_O (3.33 ppm).

**Figure 6 molecules-29-04536-f006:**
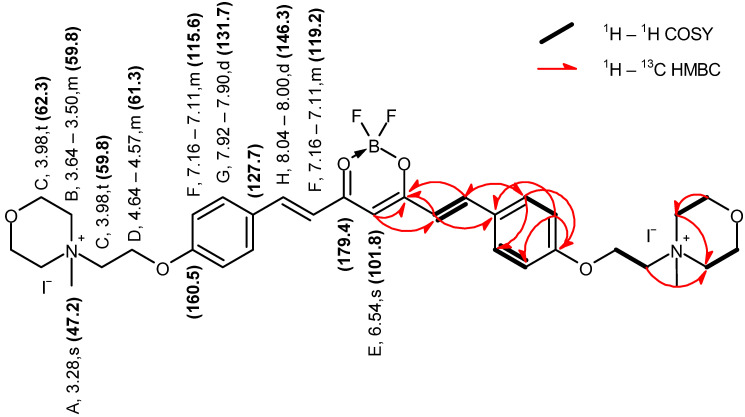
Graphical representation of the key NMR correlations (^1^H–^1^H COSY and ^1^H–^13^C HMBC) of compound **3** in DMSO–*d*_6_. The values of chemical shifts are expressed in ppm.

**Figure 7 molecules-29-04536-f007:**
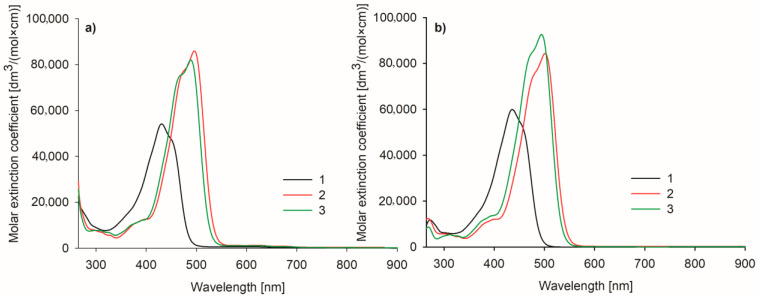
Spectra of the studied curcuminoids **1**–**3** measured in (**a**) DMF and (**b**) DMSO.

**Figure 8 molecules-29-04536-f008:**
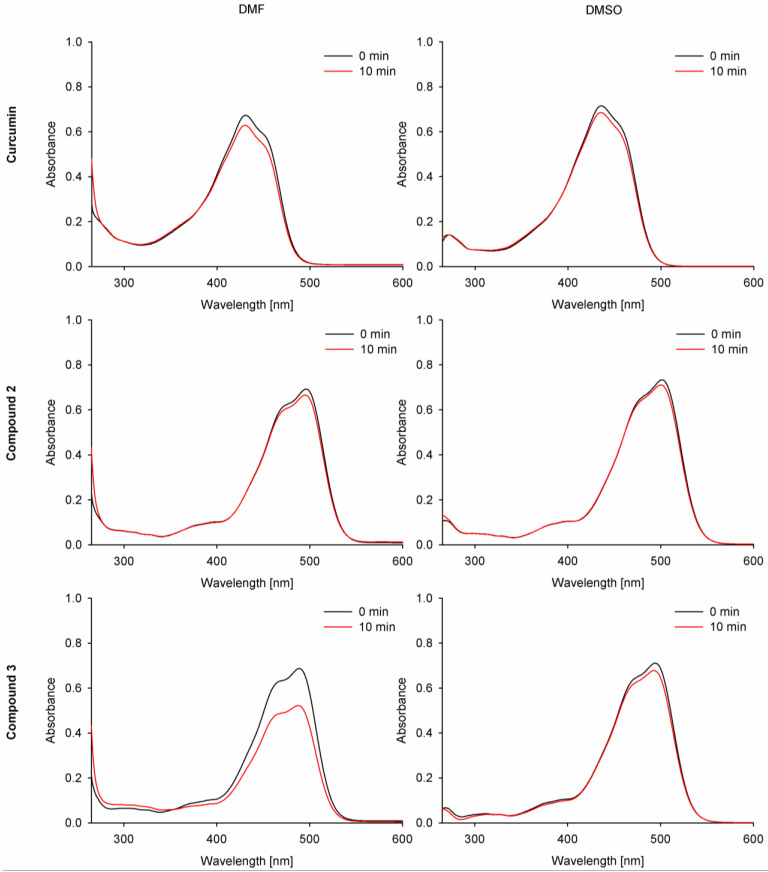
Photodegradation spectra of the studied compounds **1**–**3** in DMF and DMSO.

**Figure 9 molecules-29-04536-f009:**
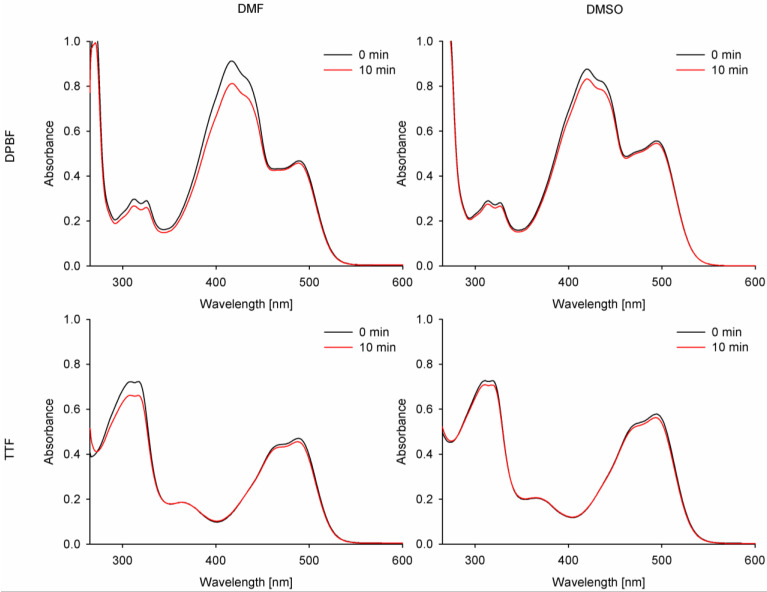
Combined UV-vis spectra before and after irradiation of compound **3** with DPBF or TTF with 489 nm (DMF) and 495 nm (DMSO) light.

**Figure 10 molecules-29-04536-f010:**
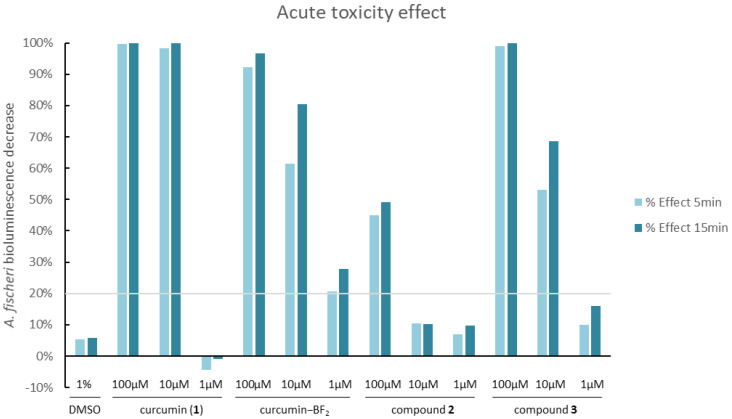
Changes in the bioluminescence of *Aliivibrio fischeri* upon exposure to different concentrations of the tested compounds, **2** and **3**, and the reference curcumin, **1**. The grey line indicates the level below which compounds are considered non-toxic (20% bioluminescence decrease). Values above the grey line can be considered toxic [48].

**Figure 11 molecules-29-04536-f011:**
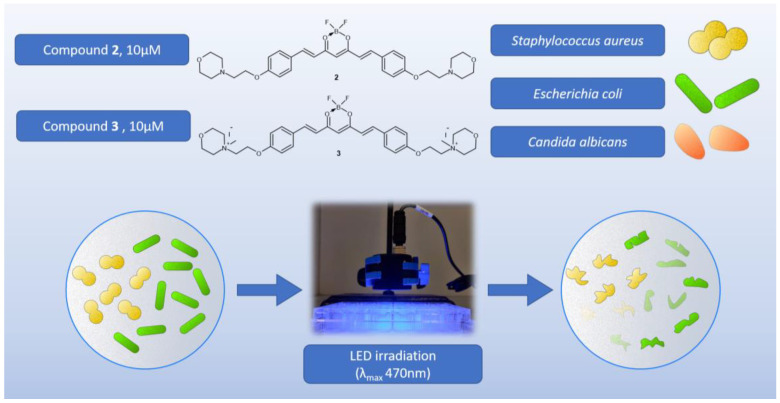
Graphical representation of the first screening experiment on microorganisms.

**Figure 12 molecules-29-04536-f012:**
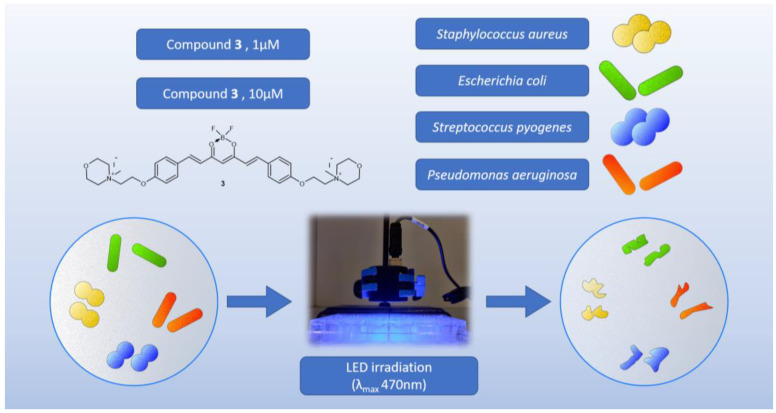
Graphical representation of the second experiment on bacteria.

**Figure 13 molecules-29-04536-f013:**
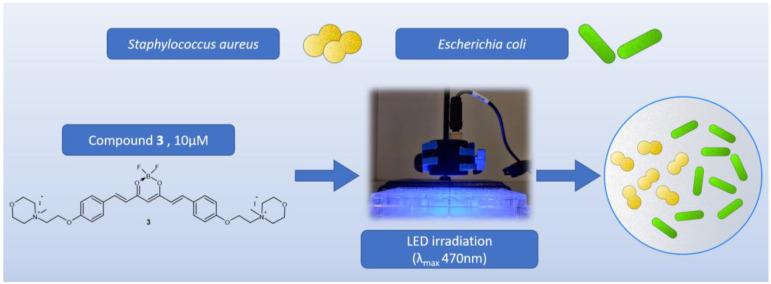
Graphical representation of the third experiment on bacteria.

**Figure 14 molecules-29-04536-f014:**
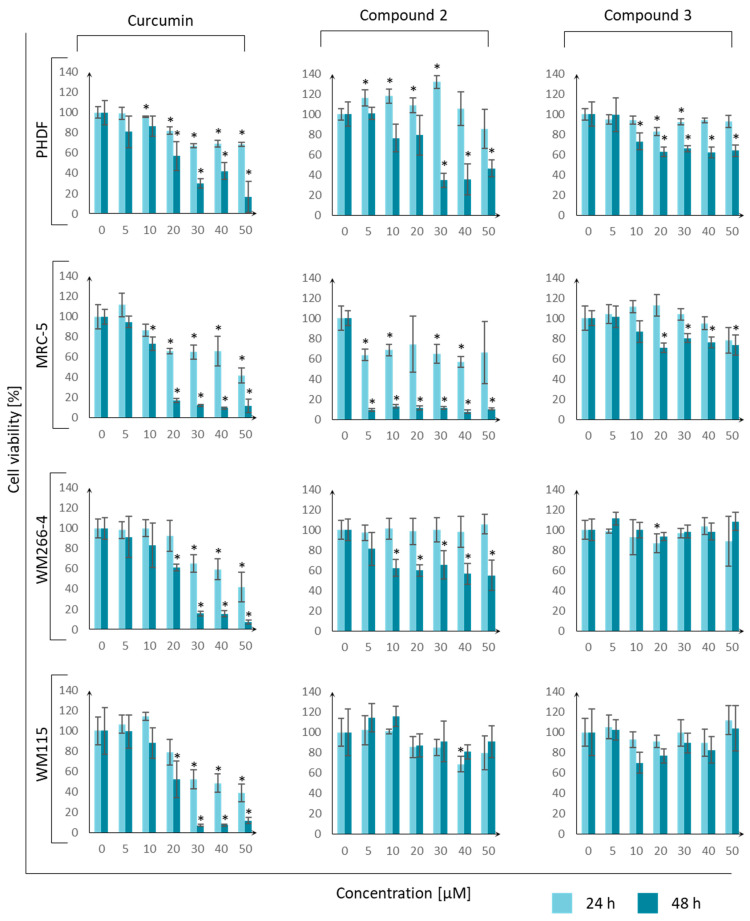
Dependence of cell viability (PHDF—primary human dermal fibroblasts, MRC-5—lung fibroblasts, WM266-4—melanoma, and WM115—melanoma) on the concentration of the tested compounds (curcumin, **2**, and **3**) measured at two time points (24 h and 48 h) during incubation. Error bars represent standard deviations. An asterisk “*” means that the value differs significantly from the control (Student’s *t*-test, *p*-value < 0.05).

**Table 1 molecules-29-04536-t001:** ^1^H and ^13^C NMR data obtained for compound **2** in DMSO–*d*_6_, including key correlations determined from ^1^H–^13^C HSQC and ^1^H–^13^C HMBC spectra.

Symbol	δ_H_ (ppm)	Multiplicity (J_H-H_ in Hz)	^1^H–^13^C HSQC δ_C_ (ppm)	^1^H–^13^C HMBC δ_C_ (ppm)
A	2.47	m	54.1	54.1, 66.6
B	2.71	t (5.7)	57.3	54.1, 66.2
C	3.58	m	66.6	54.1, 66.6
D	4.19	t (5.7)	66.2	162.3, 57.3
E	6.50	s	102.1	119.2, 179.6
F	7.09–7.07	dd (12.3, 3.4)	115.8, 119.2	102.1, 115.8, 127.4, 162.3, 179.6
G	7.86–7.83	d (8.9)	132.2	132.2, 146.8, 162.3
H	8.00–7.96	d (15.6)	146.8	119.2, 127.4, 132.2, 179.6

**Table 2 molecules-29-04536-t002:** ^1^H and ^13^C NMR data obtained for compound **3** in DMSO–*d*_6_, including key correlations determined from ^1^H–^13^C HSQC and ^1^H–^13^C HMBC spectra.

Symbol	δ_H_ (ppm)	Multiplicity (J_H-H_ in Hz)	^1^H–^13^C HSQC δ_C_ (ppm)	^1^H–^13^C HMBC δ_C_ (ppm)
A	3.28	s	47.2	59.8, 62.3
B	3.64–3.50	m	59.8	59.8
C	3.98	t (4.7)	59.8, 62.3	59.8
D	4.64–4.57	m	61.3	
E	6.54	s	101.8	119.2, 179.6
F	7.16–7.11	m	115.6, 119.2	115.6, 127.7, 160.5, 179.4
G	7.92–7.90	d (8.9)	131.7	131.7, 146.3, 160.5
H	8.04–8.00	d (15.6)	146.3	131.7, 179.4

**Table 3 molecules-29-04536-t003:** Photostability of the studied compounds when irradiated with visible light over 450 nm.

	10^−5^ Φ_p_	
Compound	DMF	DMSO
**2**	2.90	5.41
**3**	20.10	6.13
**Curcumin**	9.99	8.31
**ZnPc**	1.02 [38]	0.35 [39]

**Table 4 molecules-29-04536-t004:** Singlet oxygen quantum yields for compounds **2** and **3** measured by an indirect method with 1,3-diphenylisobenzofuran, DPBF, and tetrathiafulvalene, TTF. ZnPc—zinc(II) phthalocyanine.

Compound	Solvent	DPBF	TTF
**2**	DMF	<0.01	<0.01
	DMSO	0.024	<0.01
**3**	DMF	0.026	0.084
	DMSO	0.046	0.057
**Curcumin**	DMF	-	0.098
	DMSO	-	0.116
**ZnPc**	DMF	0.56 [39]	
	DMSO	0.67 [39]	

**Table 5 molecules-29-04536-t005:** Log_10_ reductions in common microbial strains *S. aureus* (S.a.), *E. coli* (E.c.), and *C. albicans* (C.a.) from the first screening experiment.

	Compound 2			Compound 3		
	S.a.	E.c.	C.a.	S.a.	E.c.	C.a.
L+P−	0.16	−0.03	−0.06	0.16	−0.03	−0.06
L−P+	0.05	0.52	0.04	−0.01	0.58	−0.09
L+P+	2.13	0.19	0.01	>5.54	>4.46	−0.04

L+/−—with/without light (irradiation); P+/−—with/without photosensitizer.

**Table 6 molecules-29-04536-t006:** Presentation of log_10_ reduction after various irradiation time points in the second experiment. Compound **3** was tested at concentrations of 10 µM and 1 µM. Maximum measured eradication at the first time point is presented in bold. Eradication of 99.9% of the bacteria (log_red_ > 3) is underlined. S.a.—*S. aureus*; S.p.—*S. pyogenes*; E.c.—*E. coli*; P.a.—*P. aeruginosa*.

	Compound 3, 10 µM—Log Red.				Compound 3, 1 µM—Log Red.			
Irradiation time	S.a.	S.p.	E.c.	P.a.	S.a.	S.p.	E.c.	P.a.
5 min	3.37	>4.93	1.15	4.29	−0.01	3.03	0.83	0.86
10 min	4.09				0.77	4.51		
15 min	>5.40		3.08	4.75	1.01	>4.93	1.57	2.32
20 min					1.42			
25 min			4.53	>4.85	1.72		1.82	2.83
30 min								3.32

**Table 7 molecules-29-04536-t007:** Table describing the cases in which 99.9% of microorganisms were eradicated (log reduction > 3) by compound **3** at the applied light dose. S.a.—*S. aureus*; S.p.—*S. pyogenes*; E.c.—*E. coli*; P.a.—*P. aeruginosa*.

Bacterial Strain	Time Point [min]	Log Reduction	Compound Concentration [µM]	Fluence [J/cm^2^]
*S.a.* G+	5	3.37	10	5
*S.p.* G+	5	4.93	10	5
	5	3.03	1	5
*E.c.* G−	15	3.08	10	15
*P.a.* G−	5	4.29	10	5
	30	3.32	1	30

**Table 8 molecules-29-04536-t008:** Presentation of the results obtained in experiment 3.

	Log N ± SD	
	*S. aureus*	*E. coli*
L−D+	6.88 ± 0.04	5.98 ± 0.41
L−D−	6.88 ± 0.02	5.69 ± 0.03

N—number of CFUs; L—sample without irradiation; D+/−—with/without photodegradation products.

**Table 9 molecules-29-04536-t009:** Presentation of the IC_50_ values calculated for the tested cell lines: PHDF—skin fibroblasts, MRC-5—lung fibroblasts, WM266-4—melanoma (low-melanotic), WM115—melanoma (high-melanotic), after 24 h and 48 h of incubation with the compounds (curcumin and compounds **2** and **3**). n—normal cells; c—cancerous cells.

Cell Line/Compound	Curcumin IC_50_ 24 h IC_50_ 48 h	Compound 2 IC_50_ 24 h IC_50_ 48 h	Compound 3 IC_50_ 24 h IC_50_ 48 h
PHDF—**n**	>50 µM 24.6 ± 11.5 µM	>50 µM 28.4 ± 11.8 µM	>50 µM >50 µM
MRC-5—**n**	39.5 ± 8.8 µM 18.1 ± 4.4 µM	>50 µM 3.8 ± 2.3 µM	>50 µM >50 µM
WM266-4—**c**	44.9 ± 10.7 µM 20.6 ± 9.1 µM	>50 µM >50 µM	>50 µM >50 µM
WM115—**c**	39.5 ± 9.4 µM 18.1 ± 11.0 µM	>50 µM >50 µM	>50 µM >50 µM

**Table 10 molecules-29-04536-t010:** Examples of photosensitizers from various chemical groups, designed for antimicrobial photodynamic therapy.

Photosensitizer	Irradiation Parameters	Strains	Efficiency	Ref.
Phtalocyanine derivative 100 µM	736 nm; 1.8 J/cm^2^; 10 min	*P. aeruginosa*	4.6–6.4 log	[13]
Sulfanyl tribenzoporphyrazines	660 nm; 3.6 J/cm^2^; 3 mW/cm^2^; 20 min			[11,54]
1 µM		*S. aureus*	4.8 log	
100 nM		*S. aureus*	3.2 log	
BODIPY derivatives	350–800 nm; 70 mW/cm^2^ at 500 nm			[55]
1 μM	5 min	*S. aureus*	>5 log	
5 μM	15 min	*E. coli*	~2.5 log up to 3.5 log with 50 mM KI	
5 μM	30 min	*C. albicans*	up to 5 log with 50 mM KI	
Pyridinium and imidazolium porphyrins	White light; 348 J/cm^2^; 20 h			[56]
1.5 µM		*S. aureus*	3 log (99.9%)	
2.5 µM		*E. coli*	3 log (99.9%)	
20 µM		*P. aeruginosa*	3 log (99.9%)	
Methylene blue	625 nm; 7 mW/cm^2^; 18 J/cm^2^			[57]
0.62 µg/mL		*S. aureus*	6 log	
10–20 µg/mL		*P. aeruginosa*	6 log	
Toluidine blue	670 nm; 97.65 J/cm^2^; 5 min			[58]
50 µg/mL		*S. aureus*	2.66 log	
		*P. aeruginosa*	2.36 log	
Rose bengal	515 nm; 5.8 mW/cm^2^			[59]
0.62 µg/mL	18 J/cm^2^	*S. aureus*	6 log	
0.31 µg/mL	37 J/cm^3^	*S. aureus*	6 log	
Curcumin 10 µM	440 nm; 1.944 J/cm^2^; 3.6 mW/cm^2^; 8.8 min	*S. aureus*	5.3 log	[60]
Zinc phthalocyanine RLP068/Cl	600–700 nm; 30 J/cm^2^; 50 mW/cm^2^; 10 min			[61]
64 ng/mL		*S. aureus*	5–6 log	
26 µg/mL		*P. aeruginosa*	5–6 log	
Compound 3	470 nm			This study
10 µM = 8.66 µg/mL	15 min; 15 J/cm^2^	*S. aureus*	>5.4 log	
	5 min; 5 J/cm^2^	*S. pyogenes*	>4.93 log	
	25 min; 25 J/cm^2^	*E. coli*	4.53 log	
	25 min; 25 J/cm^2^	*P. aeruginosa*	>4.85 log	

## Data Availability

The raw data supporting the conclusions of this article will be made available by the authors on request.

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
