# Peer review of "Quaternized Curcumin Derivative—Synthesis, Physicochemical Characteristics, and Photocytotoxicity, Including Antibacterial Activity after Irradiation with Blue Light"

_molecules, 2024, doi:10.3390/molecules29194536_

Round 1
Reviewer 1 Report
Comments and Suggestions for Authors
This paper represents a very complex study combining the assessment of the biological activity of curcumin coupled with quaternary ammonium compounds and their potential in the antimicrobial photodynamic therapy that shows a promising disinfecting potential. The study takes advantage of a novel approach towards the synthesis of complexes carrying a biomolecule, which is gaining attention form the scientific community, and is supported by a vast array of new data.
I do have several questions and/or recommendations, primarily connected to the methodology:
- First of all, the experimental design is immensely complex, as it comprises the synthesis and physicochemical and biological characterization of 3 complexes, in vitro assays done on microorganisms (or planktonic cells?) and four eukaryotic cell lines. This paper desperately needs a figure that will display the methodology in a comprehensive manner as the reader may get lost quickly in all the experiments that had been done.
- The manufacturers of the chemicals lack city and country of origin.
- Based on what rationale were the curcumin doses selected for each experiments? The concentration range changes amongst cell models, and it is not clear why.
- Although the methods are described in detail, for the most part, the instruments which were used are missing. Please provide the appropriate equipment for each experimental step with the manufacturer’s information for reproducibility.
- How was the zield and puritz of each synthesized complex calculated?
- Whilst the authors state the origin of the PHDF cells, the origin of the remaining cell lines is missing. What media and conditions were used for the cell incubation? If the PHDF cells came straight form patients, the obtention of the patient consent must be indicated in the subsection of the manuscript (informed consent and IRB approval). Also the IRB approval number is missing.
- I am surprised that no statistical analysis was done, particularly in case of the cell studies where controls were present, hence a comparative analysis could be done. The authors do mention statistics at some point of the paper (Student’s t-distribution value), however no further details are provided.
The Discussion is interesting however I am curious as to any limitations that may have affected the outcomes of this study.
On a final note, curcumin is known for its antioxidant properties. If the principle of antimicrobial photodynamic therapy is the generation of ROS, is I possible that curcumin could interfere with such technology by scavenging (for example) the singlet oxygen that is being produced?
Author Response
Reviewer 1
This paper represents a very complex study combining the assessment of the biological activity of curcumin coupled with quaternary ammonium compounds and their potential in the antimicrobial photodynamic therapy that shows a promising disinfecting potential. The study takes advantage of a novel approach towards the synthesis of complexes carrying a biomolecule, which is gaining attention form the scientific community, and is supported by a vast array of new data.
I do have several questions and/or recommendations, primarily connected to the methodology:
Thank you very much for all your recommendations and insightful comments.
- First of all, the experimental design is immensely complex, as it comprises the synthesis and physicochemical and biological characterization of 3 complexes, in vitro assays done on microorganisms (or planktonic cells?) and four eukaryotic cell lines. This paper desperately needs a figure that will display the methodology in a comprehensive manner as the reader may get lost quickly in all the experiments that had been done.
Thank you very much for this advice. We added Figures 11, 12 and 13 describing methodology according to your suggestion.
Microorganisms can be tested in two ways: either in the form of a biofilm or in a planktonic form (bacterial suspension). In our case, the tests were conducted on a planktonic form of microbes.
- The manufacturers of the chemicals lack city and country of origin.
Thank you very much for highlighting the issue. Whenever possible, the city and country were added.
- Based on what rationale were the curcumin doses selected for each experiments? The concentration range changes amongst cell models, and it is not clear why.
Thank you for the comment and the question. In our studies conducted so far (https://doi.org/10.1007/s00044-024-03233-z), we have shown that compounds from the curcuminoid group exhibit biological activity against human cell lines at low micromolar concentrations. However, different doses of compounds are required to demonstrate biological activity in relation to different cell lines. In this study, we have tested a wide range of concentrations to determine the active and potential therapeutically applicable concentrations and to calculate the IC50 values. The range was 0-50 µM. The concentrations selected for aPDT experiments are based on our expertise and previous experiments performed with structurally similar compounds. Each time we started with 10 µM, and after activity confirmation, we lowered the concentration to 1 µM in order to check if this activity depended on the concentration used. In the case of Microtox assay, the test is usually done in the same way and the concentrations that we used were 1, 10, and 100 µM.
- Although the methods are described in detail, for the most part, the instruments which were used are missing. Please provide the appropriate equipment for each experimental step with the manufacturer’s information for reproducibility.
All biological experiments were performed using the same equipment, so to avoid duplication of data and artificial extension of the article we decided to indicate this fact in one place in the text. This equipment is described in the Materials and methods section, precisely section 4.3.2. In-vitro photodynamic antimicrobial activity:
“Light parameters - Lamp type – LED λmax ~ 470 nm (LIU470A LED Array Light Source Thorlabs Inc., Bergkirchen, German), irradiance – 16.67 mW/cm2 (measured with PM16-130 Power Meter Thorlabs Inc., Bergkirchen, Germany and Optel Opole Sp. z o.o., Opole, Poland).”
- How was the zield and puritz of each synthesized complex calculated?
To calculate the yield, we first calculated the expected mass of the compound in milligrams. This was possible given that we knew the exact amount of reagents used in the reaction and stoichiometry. Additionally, we weighed the product after the reaction was completed and the substance was purified. With this data, we used the proportion: expected mass [mg] - 100% yield; obtained mass [mg] - x% yield. Purity was carefully assessed based on the methods described in the experimental section (4.1.3. Synthesis of 2 via aldol condensation – stage III, 4.1.4. Synthesis of compound 3 – stage IV).
- Whilst the authors state the origin of the PHDF cells, the origin of the remaining cell lines is missing. What media and conditions were used for the cell incubation? If the PHDF cells came straight form patients, the obtention of the patient consent must be indicated in the subsection of the manuscript (informed consent and IRB approval). Also the IRB approval number is missing.
Thank you for the comment. The origin of the PHDF cells was highlighted in the text "PHDF cells were extracted from skin pieces collected following mastectomy procedures". For culturing the PHDF cell line the medium contains Dulbecco′s Modified Eagle′s Medium (DMEM) (Biowest, France), 10% of fetal bovine serum (Biowest, France), and 1% of penicillin/streptomycin agents (Merck Millipore Corporation, Germany). Cells were incubated in a humidified incubator containing 5% CO2. The PHDF cell line was isolated and used in the work published previously (Cells 2021, 10(8), 1933; https://doi.org/10.3390/cells10081933). The study was conducted according to the guidelines of the Declaration of Helsinki, and approved by the Bioethical Commission of the Poznan University of Medical Sciences, Poznan, Poland (protocol code #816/10; 17 June 2010), and the Local Ethics Committee for Animal Experimentation, PoznaÅ„, Poland (protocol code #74/2010; 8 October 2010). Detailed information about the PHDF cell line was added to the methodology part of the manuscript.
- I am surprised that no statistical analysis was done, particularly in case of the cell studies where controls were present, hence a comparative analysis could be done. The authors do mention statistics at some point of the paper (Student’s t-distribution value), however no further details are provided.
Thank you for this observation. Statistical calculations were added to Figure 14.
- The Discussion is interesting however I am curious as to any limitations that may have affected the outcomes of this study.
PACT/APDT studies have several limitations. For example, the light source. Our ThorLabs LED lamp used for biological studies was placed 1 cm above the samples. There was a risk that the observed effect was partly due to the increase in the temperature of the bacterial suspension. However, we performed tests that proved that the increase in the temperature of the bacterial suspension was not observed. In our publication, only in vitro studies were conducted. In vivo, study results may differ dramatically because we do not know the organ toxicity or distribution after administration or the pharmacokinetic and pharmacodynamic properties. Additionally, blue light has low tissue penetration, so it is only suitable for superficial infections.
- On a final note, curcumin is known for its antioxidant properties. If the principle of antimicrobial photodynamic therapy is the generation of ROS, is I possible that curcumin could interfere with such technology by scavenging (for example) the singlet oxygen that is being produced?
That’s a very interesting question. In our opinion, when we discuss curcumin alone, such risk is real. But in the case of the complexes/derivatives 2 and 3, it deteriorates as the mentioned antioxidant properties of curcumin rely on the presence of the free diketo moiety and free hydroxyl groups that are not present in the derivatives 2 and 3. Diketo moiety is blocked with BF2 complex and hydroxyls (-OH) are blocked with the alkyl chain with morpholine. So in the case of compounds 2 and 3, we shouldn’t expect a high level of antioxidant properties. As such an effect might impact the aPDT efficacy, further research on these compounds will be expanded by the determination of the antioxidant properties, e.g. by using the DPPH assay and will be the subject of our future papers.

Reviewer 2 Report
Comments and Suggestions for Authors
This manuscript describes the synthesis, physicochemical characteristics and photocytotoxicity of a curcumin with quaternized morpholino nitrogen atoms, including antibacterial activity after irradiation with blue light. My overall impression is that the manuscript should become suitable for publication after some minor revisions. I have some concerns about the long irradiation times that have been used and relatively high IC50 values that are reported. A table should be added comparing the aPDT data for 3 with other quaternary ammonium compounds that have been studied for other types of molecular dyes, such as porphyrins, phthalocyanines and BODIPYs to make it clear that the results reported are not particularly promising from these standpoints. A reasonable conclusion is that this study demonstrates that curcumins of this type potentially merit further study but additional structural modification steps will be required to further enhance uptake and the singlet oxygen photosensitizer properties. Arguably, this should already have been evident at the outset, but the data set provided will still be of interest to researchers active in the field despite the limited range of curcumin dyes involved.
I urge the authors to also consider the following less important points when preparing their revised manuscript:
(a) It should probably be stated in the title line of Table 4 that singlet oxygen quantum yield values are involved. An extra column for reference numbers would be desirable where the ZnPc values are concerned given these are from the literature.
(b) Figures 7-9 need to be redrafted. Different more legible text fonts would be desirable. A decimal point should be the placeholder rather than a comma in the spectral labels. Thicker linewidths should be used for the axes and spectral data. Why are there vertical black lines to the left and horizontal black lines at the top of some of these figures?
(c) The structure in the top left corner of page 5 should be removed.
(d) Can the NMR data in Figure 5 be digitized and replotted?
(e) The significance of the horizontal gray line in Figure 10 shouldbe described in the caption.
(f) Axes should probably be added to the plots in Figure 11. The significance of the error bars should be described briefly in the caption.
Comments on the Quality of English LanguageEnglish language quality is acceptable. Just needs a careful proofread at this point.
Author Response
Reviewer 2
This manuscript describes the synthesis, physicochemical characteristics and photocytotoxicity of a curcumin with quaternized morpholino nitrogen atoms, including antibacterial activity after irradiation with blue light. My overall impression is that the manuscript should become suitable for publication after some minor revisions. I have some concerns about the long irradiation times that have been used and relatively high IC50 values that are reported. A table should be added comparing the aPDT data for 3 with other quaternary ammonium compounds that have been studied for other types of molecular dyes, such as porphyrins, phthalocyanines and BODIPYs to make it clear that the results reported are not particularly promising from these standpoints. A reasonable conclusion is that this study demonstrates that curcumins of this type potentially merit further study but additional structural modification steps will be required to further enhance uptake and the singlet oxygen photosensitizer properties. Arguably, this should already have been evident at the outset, but the data set provided will still be of interest to researchers active in the field despite the limited range of curcumin dyes involved.
Thank you very much for your opinion about our manuscript. Appropriate Table 10 was prepared and added to the text.
I urge the authors to also consider the following less important points when preparing their revised manuscript:
(a) It should probably be stated in the title line of Table 4 that singlet oxygen quantum yield values are involved. An extra column for reference numbers would be desirable where the ZnPc values are concerned given these are from the literature.
Thank you for the suggestion. Changes were made accordingly.
(b) Figures 7-9 need to be redrafted. Different more legible text fonts would be desirable. A decimal point should be the placeholder rather than a comma in the spectral labels. Thicker linewidths should be used for the axes and spectral data. Why are there vertical black lines to the left and horizontal black lines at the top of some of these figures?
Thank you for your suggestion, the fonts and line widths have been enlarged. The additional lines were deleted, they were supposed to match the remaining figures, although it seems that their appearance was unnecessary and altered the clarity of the figures.
(c) The structure in the top left corner of page 5 should be removed.
There is no structure in the top left corner of page 5.
(d) Can the NMR data in Figure 5 be digitized and replotted?
The figures were prepared in accordance with the journal’s recommendations. Original fid files are available from the authors on request.
(e) The significance of the horizontal gray line in Figure 10 should be described in the caption.
Thank you for the suggestion. Changes were made accordingly.
(f) Axes should probably be added to the plots in Figure 11. The significance of the error bars should be described briefly in the caption.
Thank you for the suggestion. Changes were made accordingly.
